# Nuclear Receptor Metabolism of Bile Acids and Xenobiotics: A Coordinated Detoxification System with Impact on Health and Diseases

**DOI:** 10.3390/ijms19113630

**Published:** 2018-11-17

**Authors:** Manon Garcia, Laura Thirouard, Lauriane Sedès, Mélusine Monrose, Hélène Holota, Françoise Caira, David H. Volle, Claude Beaudoin

**Affiliations:** 1Université Clermont Auvergne, GReD, CNRS UMR6293, INSERM U1103, 28, Place Henri Dunant, BP38, F63001 Clermont-Ferrand, France; manon.garcia@uca.fr (M.G.); laura.thirouard@uca.fr (L.T.); laurianesedes@hotmail.fr (L.S.); melusine.monrose@uca.fr (M.M.); helene.holota@uca.fr (H.H.); francoise.caira@uca.fr (F.C.); 2Centre de Recherche en Nutrition Humaine d’Auvergne, 58 Boulevard Montalembert, F-63009 Clermont-Ferrand, France

**Keywords:** FXR, PXR, CAR, bile acids, xenobiotics, metabolism and transport, cancer, drug resistance

## Abstract

Structural and functional studies have provided numerous insights over the past years on how members of the nuclear hormone receptor superfamily tightly regulate the expression of drug-metabolizing enzymes and transporters. Besides the role of the farnesoid X receptor (FXR) in the transcriptional control of bile acid transport and metabolism, this review provides an overview on how this metabolic sensor prevents the accumulation of toxic byproducts derived from endogenous metabolites, as well as of exogenous chemicals, in coordination with the pregnane X receptor (PXR) and the constitutive androstane receptor (CAR). Decrypting this network should provide cues to better understand how these metabolic nuclear receptors participate in physiologic and pathologic processes with potential validation as therapeutic targets in human disabilities and cancers.

## 1. Introduction

Nuclear receptors (NRs) are ligand-dependent transcription factors that regulate a variety of physiological processes by inducing the transcription of target genes. Among the members of this large NRs superfamily, the metabolic receptors farnesoid X receptor α (FXRα, also known as NR1H4), pregnane X receptor (PXR, called steroid X receptor SXR or NR1I2), and constitutive androstane receptor (CAR, also named NR1I3) have been intensively studied for their ability to regulate energy metabolism by affecting fatty acid, lipid, and glucose homeostasis (for recent reviews, see [1,2,3]). As sensors of endogenous and exogenous chemicals, these three NRs have also been largely explored for their capacity to regulate the expression of transporters and drug-metabolizing enzymes as adaptive responses to prevent the accumulation of toxic chemicals in the body. Like most others NRs, these receptors have a characteristic modular structure which includes a C-terminal ligand-binding domain (LBD) and a N-terminal highly conserved DNA-binding domain (DBD) that allows them to bind to specific nucleotide consensus sequences, hereafter referred to as *DNA response elements* (Figure 1A). As common (non)permissive heterodomeric partners, retinoid X receptors (RXRα/β/γ, also known as NR2B1/NR2B2/NR2B3) are required for high affinity binding to DNA, and despite the fact that RXRα remains the predominant isotype in many tissues, all three RXR subtypes generally make these NRs competent for chromatin binding and transcriptional regulation of target gene expression in concert with corepressor complexes or recruited co-activator proteins (Figure 1B and [4,5]).

The discovery of bile acids, bilirubin, and xenobiotics, including pharmaceutical drugs and environmental pollutants, as specific ligands for these NRs and the assignment of their regulatory function to hepatobiliary transporter genes have given the way to an extensive area of research over recent decades. Subsequently, a rapidly growing number of publications have dealt with the regulatory mechanisms of enterohepatic bile acids (BAs) circulation in health and diseases with a clear emphasize on therapeutical strategies for metabolic disabilities and cancers. The aim of the present review is to describe the most relevant functions of FXRα, PXR, and CAR, with a focus on convergence points of crosstalks that orchestrate detoxification pathways of endogenous and exogenous chemicals that may also adversely affect male reproductive function and increase the risk of developing a testis tumorigenic phenotype accompanied by drug resistance.

## 2. Structure-Activity Relationship of Bile Acids and Bile Acid Derivatives in Regards to Sensing Nuclear Receptors

### 2.1. Bile Acids as Endogenous Ligands for Metabolic Nuclear Receptors

Actively synthesized from cholesterol in the liver, bile acids (BAs) are amphipathic molecules with detergent-like properties that are essential for hepatic bile formation, cholesterol elimination, and the absorption of dietary lipids from the small intestine. In humans, the majority of BA synthesis in the hepatocytes occurs through a classical pathway that is initiated by the rate-limiting enzyme cholesterol 7α-hydroxylase (CYP7A1). This major classical (neutral) pathway forms the primary BAs, referred to as cholic acid (CA) and chenodeoxycholic acid (CDCA), following a multistep enzymatic process involving the key sterol 12α-hydroxylase (CYP8B1) and sterol-27 hydroxylase (CYP27A1) enzymes respectively. In complement, CYP27A1 initiates an alternative (acidic) pathway of BA synthesis by hydroxylation and oxidation of cholesterol that is further hydroxylated by oxysterol 7α-hydroxylase (CYP7B1) to CDCA. Shortly after their synthesis, BAs are conjugated to amino acids glycine and taurine to form sodium salts (bile salts), and excreted into the gallbladder, before being released into the intestinal tract where microbial transformations in the gut convert the primary BAs CA and CDCA to corresponding secondary BAs deoxycholic acid (DCA) and lithocholic acid (LCA). Most BAs listed in Table 1, including the CDCA-derived α-muricholic acid (αMCA) and β-muricholic acid (βMCA) in mice, are reabsorbed in the ileum and transported back to the liver via the portal vein (recently reviewed in [6]).

Adequate concentrations of BAs in the intestinal lumen for digestion and their efficient reabsorption in the terminal ileum ensure the maintenance of circulating BA pools within the enterohepatic circulation. Indeed, BAs undergo enterohepatic circulation, with about 95% being reabsorbed by active transport in the ileal enterocytes and recycled back to the liver through the portal vein for further secretion into the biliary system and gallbladder. In each enterohepatic cycle, hepatocytes must synthesize de novo BAs through a tightly regulated process to replace those that are eliminated in the feces after escaping the ileum. In addition to the well described physiochemical characteristics of these natural detergents in the maintenance of cholesterol homeostasis, biliary excretion of hydrophobic compounds, and intestinal fat absorption, clear evidence also indicates that high or abnormal BA concentrations may become cytotoxic to cells. This is particularly the case for both hepatocytes and enterocytes, as well as for cholangiocytes (biliary epithelial cells), with an increased risk of digestive system diseases including cholestasis and gastrointestinal carcinogenesis (reviewed in [7,8,9]). Obviously, it turns out that cells exposed to BAs protect themselves too by tightly regulating metabolic integrators and biological processes in order to control BA biosynthesis, metabolism, and transport through ligand activation of NRs including FXRα, and the promiscuous xenobiotic receptors PXR and CAR that respond to a larger variety of structurally-unrelated exogenous compounds [10,11,12,13]. In short, CDCA is the most efficacious ligand of FXRα, followed by LCA, DCA, and CA, whereas the secondary BA LCA and its oxidized 3-keto form is the main, if not the only, potent activator of PXR. As with it closest relative PXR, CAR binds a broad variety of structurally diverse molecules counting CA and 6-keto-LCA [14]. In complement to the vitamin D receptor (VDR also known as NR1I1) and the cell-surface receptor TGR5 (also known as GPBAR1; G-protein coupled bile acid receptor 1), which are outside of the scope of this review [15,16], these three NRs exert their functions as sensors of BA levels, and elicit transcriptional alterations in order to regulate BA homeostasis, as discussed below.

### 2.2. Molecular Biology of Bile Acid/Xenobiotic Nuclear Receptors

FXRα was the first identified BA-activated nuclear receptor and the most dedicated NR to BA signaling [10]. Two distinct *FXR* genes (*FXRα*, *NR1H4*; *FXRβ*, *NR1H5*) are evolutionarily conserved between human and rodent; but in contrast to the mouse, *FXRβ* constitutes a pseudogene in human [17,18]. Using alternative promoters and splicing, four functional variants have been described for FXRα (α1–4), with differences in both tissue- and species-specific expression as well as in ligand-dependent transcriptional activities [19,20]. Largely expressed in tissues engaged in the human enterohepatic circulation, FXRα variants are all present in the small intestine, in contrast to the liver and adrenals, which express mainly FXRα1 and α2, while FXRα3 and α4 are predominant in the colon and kidney. The situation is slightly different in mouse because liver and small intestine express all variants, whereas kidney and stomach express only FXRα3 and α4 [21]. Yet, all forms of FXRα, hereafter referred to as FXRα, function as homeotrophic sensors of BAs and regulate the activity of numerous genes encoding for proteins involved in BA synthesis, transport, conjugation, and excretion. To do so, FXRα binds to a FXR response element (FXRE) as a heterodimer with RXRα, or as a monomer to regulate gene expression. As for the other NRs, two zinc finger motifs allow FXRα to interact with its response element which contains two copies of a consensus AGGTCA-like DNA sequence reiterated directly (DR), inverted (IR), or everted (ER) with various inter half-site spacing (Figure 1B). Even though the FXRα/RXRα heterodimer binds to the consensus IR-1 (IR spaced by 1 base pair) sequence with the highest affinity, this complex also binds to and activates a variety of other FXREs in conjunction with transcriptional co-activators or co-repressors that coordinate either gene activation or repression following post-translational modifications of histones and non-histone proteins. Overviews of FXRα structure, activation, interaction with RXRα, and dynamically-controlled target gene regulation have been published previously [22,23,24,25].

In complement, PXR and CAR xenosensors also play important roles in the regulation of BA metabolism and detoxication by inducing genes involved in BA conjugation and transport in order to enhance their elimination [26]. Both NRs, best known for their role in phase I (hydroxylation), phase II (conjugation) and phase III (transport) xenobiotic elimination, are mainly expressed in liver and intestine, where they likely regulate gene expression of ATP-binding cassette (ABC) transporters and of drug-metabolizing enzymes within transcriptionally-permissive RXRα heterodimers [27,28]. In human and mouse, several spliced and polymorphic variants have been described for PXR and CAR, resulting in protein isoforms that could vary in structure-function properties, as well as in either constitutive or ligand-inducible transcriptional capacities (for a complete review, see [27]). Even if these xenobiotic receptors are closely related, their moderately conserved LBD allow them to recognize a wide range of structurally-unrelated endogenous and exogenous ligands [29,30]. This may also reflect the different species ligand-binding properties of PXR and CAR described so far, although it is known that both receptors share ligands that act as dual modulators for the expression of a common set of genes [31,32,33]. Nevertheless, obligate associations of PXR and CAR with the heterodimeric partner RXRα activate specific DNA sequences that differ from one another by the number of base pairs separating the direct or everted DNA repeats. The best described CAR response elements are the phenobarbital-responsive enhancer module (PBREM) and the xenobiotic responsive enhancer module (XREM), which are found in the promoter regions of CAR target genes. Because these binding sites are shared with PXR, both NRs co-regulate a subset of their target genes despite the fact that they also separately regulate other specific genes. Inspections of these response elements indicate that both DR-3 and ER-6 type AGGTCA-like binding motifs are typically bound by PXR/RXR heterodimers, in contrast to CAR/RXR, that binds DR-4, DR-5 or ER-8 repeat elements [34]. Despite a lower affinity for the DNA consensus sequences, monomeric CAR is also described to be competent for docking onto octameric AGAGTTCA DNA motifs and for the interaction with co-activators [35,36,37,38]. Owing to their ligand-binding pocket that accommodates molecules of various shapes and sizes, it is not surprising that PXR and CAR regulate an overlapping set of target genes in BA metabolism and elimination following agonistic and antagonistic recruitment of co-activators and co-repressors at XREM regulatory regions of target genes.

### 2.3. Bile Acid/Xenobiotic Nuclear Receptors Regulation of Bile Acid Synthesis and Transport

One of the key functions of FXRα is to critically regulate BA synthesis and BA enterohepathic circulation. Mice lacking Fxrα (*Fxrα^−/−^*) were phenotypically normal but showed elevated expression of *Cyp7a1* mRNA and enlarged BA pool sizes [39,40]. By inducing the small heterodimer partner (SHP also known as NR0B2) in liver, FXRα slows down the classical biosynthetic pathway of BAs by reducing expression of *CYP7A1/Cyp7a1* and *CYP8B1/Cyp8b1* gene-encoding enzymes in humans and mice (Figure 2). This occurs in part through dimerization of SHP with liver receptor homolog-1 (LRH-1) and hepatocyte nuclear factor 4α (HNF4α) transcription factors, and repression of their binding capacities to *CYP7A1* and *CYP8B1* gene promoters, with an additional inhibitory feedback loop on *SHP* itself [41,42,43,44]. In addition, FXRα activation in the intestine stimulates expression and secretion of the fibroblast growth factor 19 (FGF19, FGF15 in rodent models) which binds FGFR4/β-Klotho receptor complexes on the cell membrane of hepatocytes after trafficking into the enterohepatic circulation [45,46,47]. This functional gut-liver signaling pathway inhibits expression of *CYP7A1* and *CYP8B1* gene products through extracellular signal-regulated protein kinase 1/2 (ERK1/2) and c-Jun N-terminal kinase 1/2 (JNK1/2) activation. In complement, experiments with liver- and ileum-specific Fxrα-deficient mice (*Fxrα^−/−^*) showed that this inhibitory metabolism of BAs dominates over the hepatic negative feedback pathway [48,49].

It has also been suggested that PXR and CAR play a role in the regulation of BA synthesis, since PXR has been described as a FXRα target gene, and its ligand activation directly regulates *SHP* expression in the HepG2 human liver cancer cell line and indirectly reduces the expression of *Cyp7a1* in mice [36,50,51]. Moreover, deletion of *CAR* results in a less severe repression of *Cyp7a1* and *Cyp8b1* after bile duct ligation in mice, indicating that CAR is partly responsible for this down-regulation when BAs accumulate in high concentrations in the liver [52]. Despite the fact that the protective effect of PXR and CAR can be explained to a certain extent by their capacity to regulate BA synthesis genes including *CYP27A1* [53], the most likely protective effects of these xenobiotic sensing NRs seem to come from their crosstalks in the management of a large set of genes that coordinately regulate BA metabolism and transport, as depicted in Figure 2.

Following their synthesis, BAs are secreted into bile, which flows through the bile duct to the intestine. Then, BAs are efficiently absorbed from the intestine, returned to the liver, and resecreted into the bile and the gallbladder in many species. In this enterohepatic circulation, BAs can promote their own biliary elimination by stimulating the hepatic apical bile salt efflux pump (BSEP; ABCB11) and the multidrug resistant-associated protein 2 (MRP2; ABCC2) expression. These two ABC transporters are rate limiting in canalicular excretion of BAs and non BA organic ions. In human and rodent, *BSEP* gene promoters are transcriptionally activated by FXRα [54,55], and mouse *Bsep* baseline-expression largely depends on the presence of this BA-activated NR, since its abundance is reduced in *Fxrα* knockout mice [40,56,57]. In contrast to BSEP, transcriptional regulation of MRP2 involves several overlapping sets of NRs, reflecting the diverse substrate spectrum of MRP2. As a matter of fact, MRP2 can mediate the excretion of a broad range of conjugated BAs and non-BA organics anions, mostly with glucuronidate and sulfate formed by phase II conjugation enzymes in hepatocytes [58]. FXRα binds with high affinity to response elements in the human and rodent *MRP2*/*Mrp2* promoters that are also shared with CAR and PXR [59]. Furthermore, FXRα increases the expression of multidrug resistance protein 3 (MDR3; ABCB4), which secretes phospholipids into the bile canaliculi, thereby promoting the formation of mixed micelles and decreasing toxicity of BAs [60]. While BAs are excreted into canalicular bile under normal conditions, most conjugated BAs are actively reabsorbed in the ileum by an apical, sodium-dependent bile salt transporter (ASBT, SLC10A2, IBAT) located on the apical side of enterocytes. In the small intestine, FXRα increases ASBT, which is important for the reabsorption of BAs from the intestinal lumen, as well as ileal bile acid binding-protein (IBABP) and organic solute transporter α/β (OSTα/β), which are both important for the transcellular transport of BAs [61,62,63]. Absorbed BAs are then transported back into portal blood by the sodium-dependent bile acid transporter (NTCP; SLC10A1) and a family of multi-specific organic anion transporters (OATPs; SLC21A) that mediate sodium-independent uptake of conjugated or unconjugated BAs, as well as bilirubin in the liver [64,65]. Regulation of *NTCP* by BAs is complex, and differs considerably among human and rodent [66]. Negative feedback inhibition of mouse and rat *Ntcp* is mediated by both FXRα/SHP-dependent and -independent mechanisms and limits hepatocellular BA uptake [67]. Similar to NTCP, FXRα represses *OATP1B1* through SHP and HNF4α and inhibits the predominant sodium-independent BA uptake system in human [68]. In contrast, OATP1B3, a multi-specific uptake system for organic anions, xenobiotics, and potentially BAs, is positively trans-activated by FXRα [69]. An alternative low level efflux of BAs into the systemic blood through the sinusoidal membrane also takes place, and is mediated by several transporters including OSTα/β, MRP3/ABCC3 and MRP4/ABCC4 [70,71]. Induction of both rodent *Mrp3* and *Mrp4* seems to be independent of FXRα, while both PXR and CAR are able to induce either *MRP3/Mrp3* or *MRP4/Mrp4* expression in human and rodent [56,72,73,74]. Collectively, a complex picture emerges where FXRα, PXR, and CAR are required to coordinate synthesis and enterohepatic cycling of BAs.

### 2.4. Regulation of Bile Acid Metabolism and Elimination by Bile Acid/Xenobiotic Nuclear Receptors

The formation of BA-conjugated metabolites was recognized early as a mechanism of BA deposition in human and other species. Shortly after their synthesis, BAs are conjugated to amino acids glycine and taurine to increase their solubility in water and to reduce their cytotoxicity [75]. Over the years, FXRα was shown not only as an essential regulator of BA conjugation enzymes such as bile acid CoA synthase (BACS) and bile acid CoA amino acid *N*-acetyltransferase (BAAT) [76,77], but also as an important factor of BA detoxification processes, since BA-activated FXRα induces *cytochrome P4503A4* (*CYP3A4*) expression and phase I hydroxylation reactions [78]. Other NRs, including PXR and CAR, have been demonstrated to activate *CYP3A4* [79,80,81,82]; besides this most characterized mechanism of gene regulation, these xenobiotic sensors are further involved in the regulation of phase II enzymes including uridine diphospho-glucuronosyltransferases (UGT), sulfotransferases (SULT) and glutathione-S-transferases (GST) [83,84]. By conjugating hydrophilic groups, UGTs, SULTs, and GSTs increase the water solubility of BAs. In human, positive regulation of *SULT2A1* gene expression by FXRα and PXR appears to play a central role in regulating BA sulfation in conjunction with CAR that is proposed to orchestrate BA sulfation and their ensuing export by the CAR-induced MRP4 basolateral export pump [56,85,86,87]. In addition to sulfation, BAs can also be detoxified through glucuronidation by the UDP-glucuronosyltransferases UGT2B4, UGT2B7, and UGT1A3, that renders BAs more water soluble and facilitates their renal elimination [88,89,90]. This combined hydroxylation/glucuronidation detoxification pathway can also be stimulated by PXR and CAR in human after UGT1A1 transcriptional activation, while BAs can modulate both UGT2B4 and UGT2B7 expression through FXRα [88,89,91]. Together with both PXR- and CAR-mediated BA metabolism, the FXRα-induced repression of basolateral uptake by NTCP and OATP, in combination with the increased expression of gene encoding proteins involved in BA detoxification (CYP3A4, UGT2Bs, SULT2A1) and in membrane transport (BSEP, MRP2/3, MRD3, OSTα/β), help to promote excretion and elimination of conjugated BAs [92,93]. It is now becoming apparent that xenobiotic receptors crosstalk with FXRα to move the metabolized and conjugated BAs and their potentially toxic metabolites into the body’s excretory pathways via the kidney or the bile, with impact, in some disease states, that emphasizes drug targeting of BA and xenobiotic-sensing NRs in precision medicine.

## 3. Modulation of Bile Acid/Xenobiotic Receptors for Therapeutic Applications

### 3.1. Modulators of Bile Acid and Xenobiotic-Sensing Nuclear Receptors

Research in the past decades has established that FXRα, along with PXR and CAR, have a crucial role in regulating all aspects of BA metabolism, and the availability of natural and non-steroidal synthetic ligands has provided important insights into the mechanisms by which NRs for BAs and xenobiotics control many diverse metabolic pathways, including glucose homeostasis and lipid metabolism. Due to broad spectrum activity and synergistic effects of these three NRs, identification of small molecules selectively targeting these receptors still represents a prolific area of research, with a large number of ligands being published in the last years and highlighted in recent comprehensive literature summaries [14,94,95,96,97]. Despite the large variety of ligands reported, so far, based on evidence from cell lines and mouse models, only the most widely-used natural and synthetic BA/xenobiotic NR ligands will be mentioned here.

Farnesol, which is a metabolite intermediate of the mevalonate biosynthetic pathway, was the first common natural FXRα ligand identified, but its weak agonist activity impedes its use [98]. Oxysterol-22(*R*)-hydroxy cholesterol, androsterone as well as 26- or 25-hydroxylated BA metabolites and poly-unsaturated fatty acids (arachidonic acid and docosahexaenoic acid) were also reported to act as weak FXRα ligands [24]. As mentioned previously, BAs exhibit distinct potencies to activate FXRα, with CDCA being the most active natural human FXRα activator, followed by DCA and LCA, but the poor selectivity of these endogenous BAs encourages the development of new synthetic and non-synthetic ligands with improved selectivity for FXRα in order to dissociate FXRα-dependent and -independent pathways. To do so, the CDCA high-affinity FXRα ligand has been subjected to intense medicinal chemistry modifications, producing several steroidal derivatives in order to generate more specific FXRα modulators which are devoid off-target effects [99,100]. Among them, the semi-synthetic BA analogue 6-ethyl-chenodeoxycholic acid (6-ECDCA, also known as INT-747, obeticholic acid, or Ocaliva/OCA) was shown to be a very potent and selective agonist in an FXRα transactivating assay, and it has become an alternative agonist to the most widely-used GW4064 FXRα ligand due to the cell-toxic effect and poor intestinal absorption of this latest non-steroidal synthetic agonist [101,102,103]. Although GW4064 has been extensively used as a molecular tool to study FXRα functions in vitro and in vivo, it was not developed further because of its liabilities, that included poor pharmacokinetic properties and the presence of the potentially toxic stilbene moiety. Many attempts to overcome these problems in combination with crystallographic studies and virtual screening campaigns have given rise to several series of derivatives, and recent progress has been made with high-affinity binding and agonist activity of structurally-diverse, natural (epigallocatechin-3-gallate, cafestol) or (semi)synthetic (GSK2324; WAY-36245; AGN29; PX20606) molecules toward FXRα. In contrast, only a limited number of FXRα antagonists (guggulsterone, stigmasterol, AGN31, AGN34) have been described in the literature [104,105]. While these latter compounds may be useful pharmacological tools to extend the understanding of FXRα functions, complementary studies are required to further validate the potency of these agents as clinic therapeutic modulators of FXRα in human disabilities.

Besides LCA, that is one of a few described natural ligands for PXR, a wide range of structurally-unrelated, endogenous and exogenous ligands were shown to selectively bind to the PXR of different species. In fact, the most well-known PXR agonists include the pharmaceutical drug rifampicin (RIF), which strongly binds to human PXR but not to mouse PXR, and the synthetic anti-glucocorticoid pregnenolone 16α-carbonitrile (PCN), that activates PXR in rodent (mouse and rat) but has no effect on human PXR [32]. These species’ selectivity in ligand-induced activation rely on clear differences into the LBD amino acid sequences of human and mouse PXR, and authorize a commensurate number of foreign compounds to occupy the large binding pocket of this receptor. This allows PXR orthologs to be activated not only by a plethora of prescription drugs (rifampicin, ketoconazole, dexametasone, acetaminophen), chemotherapeutics (cyclophosphamide, paclitaxel, taxol), steroids (progesterone, ethinylestradiol), and nutritional compounds (flavonoids), but also by a variety of environmental contaminants such as bisphenol analogs (BPA), polychlorinated biphenyls (PCBs), diethylhexyl phthalates (DHEP), and organochloride pesticides (trans-nonachlor, chlordane). Comprehensive lists of compounds affecting the biological activity of human and rodent PXR have already been presented in references [14,27,95], with an overlap in the endogenous and exogenous compounds that bind to CAR variants and orthologs.

In contrast to FXRα and PXR, CAR has been initially described as a constitutively-active NR in the absence of any activating ligand, but it has since been reported to be regulated by many endobiotics, including steroids (androstanes, estrogens, and progestins) and BA metabolites. Also, clinical drugs, pesticides, food-derived flavonoids, and alcohol-derived polyphenols are efficient CAR modulators (reviewed in [106]), establishing CAR as a critical effector of xenobiotic function and cellular toxicity. Multiple compounds with direct or indirect effects on CAR-dependent gene regulation have been discovered, but few are selective for CAR specifically, as most also bind to PXR [27,95]. Still, phenobarbital (PB) is the most potent known indirect activator of CAR that causes the receptor to undergo nuclear translocation, and exerts transcriptional activity after binding PBREM without interacting with the ligand binding domain of CAR [107]. This results from inhibition of the epidermal growth factor (EGF) receptor signaling pathway by PB, and later, CAR dephosphorylation, that activates nuclear translocation and transcriptional activity of the receptor, as reported for more recently-investigated diet flavonoids [108]. In addition, CAR activation can also be mediated by direct activators according to its remarkable species selectivity for ligand binding and activation profile. The potent ligand activators TCPOBOP (1,4-Bis[2-(3,5-dichloropyridyloxy)])benzene and CITCO (6-(4-chloropheny)imidazo[2,1-b][1,3]thiazole-5-carbaldehyde-*O*-(3,4-dichlorobenzyl)oxime) behave as direct and specific agonists for the mouse and human CAR receptors respectively [109,110]. In both cases, direct ligand-dependent activation of CAR seems to rely on the nuclear translocation step, whereby once inside the nucleus, ligand-bound CAR adopts a conformation similar to that of the constitutively-active CAR, and maintains transcriptional activity. Interestingly, the constitutive activity of CAR can be reversed, likely by a class of compounds known as inverse agonists which derive from androstane metabolites. By promoting association of weak corepressor complexes following displacement of coactivator proteins, androstanol (5α-androstan-3α-ol) and androstenol (5α-androst-16-en-3α-ol) reverse constitutive ligand-independent activity of mouse CAR [111,112,113]. A panel of structurally-diverse small molecules has been reported as being human CAR inverse agonists [114], providing novel insights into the nature of CAR regulation and function under specific physiologic or pathologic conditions, as briefly addressed in the following section.

### 3.2. Potential Clinical Use of Bile Acid and Xenobiotic Nuclear Receptor Modulators

Initial studies performed more than fifty years ago revealed that BAs act as key agents for cholesterol solubilization and removal out of the body, and for maintaining the driving force for bile flow. On the basis of these observations, oral administration of CDCA was thus used to dissolve cholesterol-gallstones in disease patients, but was replaced later by its more hydrophilic 7-epimer derivative, ursodeoxycholic acid (UDCA or Ursodiol), in order to reduce biliary cirrhosis and hepatotoxicity [115,116,117]. In contrast to CDCA, which is the most active natural human FXRα activator, UDCA, which is a naturally-occurring secondary BA in human, is devoid of any activity on FXRα in transactivating assay, even if it is reported to act as a weak PXR agonist [14,118]. Yet, UDCA and its 24-nor-UDCA synthetic analog are used to enhance impaired bile flow and excretion of BAs or others potentially toxic biliary constituents in patients with cholestatic liver diseases, due to mechanical obstruction of bile ducts attributable to tumors and gallstones, or even to metabolic disorders in BA formation and transport consecutive to hereditary genetic defects, pregnancy, or medication [103]. Repression of *Cyp7a1* and modification of the BA export pumps *Bsep* (*Abcb11*) and *Mrp2* (*Abcc2*) have been reported in animal models treated with UDCA, and likely explain its choleretic effect that relies on complex signaling networks (reviewed in [119,120]). Indeed, UDCA was not only demonstrated to improve cholestasis by reducing BA uptake and synthesis, but also by stimulating hepatocellular and biliary ductular export and elimination, making it attractive as a first-line treatment for a multitude of other cholestatic injuries, including primary biliary cirrhosis (PBC), primary sclerosing cholangitis (PSC), intrahepatic cholestasis of pregnancy (ICP), and progressive familial intrahepatic cholestasis (PFIC). However, the limited efficacy of UDCA in these cholestatic conditions has given rise to demand for the development of novel therapeutic approaches, including BA derivatives and other FXRα agonists. Indeed, experimental studies using animal models mimicking cholestatic liver injuries (bile duct ligation, genetic engineering, BA feeding) have strongly contributed to highlighting the complex interplay of BA and bilirubin-activated NRs (mainly FXRα, PXR and CAR) that orchestrates an adaptive response when intrahepatic and systemic BA levels rise (for reviews, see references [121,122]).

Due to its role in repressing BA uptake transporters (i.e., NTCP and ASBT) and inducing BA export pumps (i.e., BSEP, MRP2, and OSTα/β) along with suppression of BA synthesis (i.e., CYP7A1), FXRα activation represents another line of developed anticholestatic strategies, with (semi)synthetic steroidal (6-ECDCA) and non-steroidal FXRα agonists (fexaramine, GW4064, PX20606) being actually investigated in (pre)clinical trials. Moreover, induction of CYP3A4, SULT2A1, and UGT2Bs, along with stimulation of biliary excretion through MDR3 (ABCB4), may also enhance BA detoxification and counteract cholesterol-gallstone after FXRα activation. Therefore, it has come as no surprise that the high affinity 6-ECDCA FXRα ligand was able to restore reduced bile flow and improve cholestasis in several preclinical animal models of cholestasis [102,123], as well as in patients with PBC [124]. According to both animal studies and clinical trials, it was noted that besides cholestasis, the FXRα agonist 6-ECDCA has also shown promise in treating metabolic disorders such as non-alcoholic fatty liver disease (NAFLD), non-alcoholic steatohepatitis (NASH), and type 2 diabetes mellitus (T2DM), which is in agreement with earlier described functions of FXRα in the regulation of lipid metabolism and glucose homeostasis [125,126,127].

Additionally, PXR and CAR activators have also been broadly studied, and some of them (e.g., RIF, PB) are actually tested clinically in liver diseases to activate BA transporters and detoxification systems [128,129,130]. The importance of these two NR xenosensors in the adaptive response to toxic BAs and other compounds that accumulate in the liver has been demonstrated in numerous knockout rodent models. For example, Pxr knockout mice are more susceptible to liver injuries caused by LCA treatment or bile duct ligation than wild-type mice and the activation of Pxr by PCN reduced toxic BA-induced liver injuries in wild-type mice, but not in Pxr knockout mice [51,52,131]. Moreover, data from studies on *Fxr*^−/−^*Pxr*^−/−^ double knockout mice showed enhanced toxicity in response to BAs. Of note, CAR activation by PB or TCPOBOP when both Fxrα and Pxr are lost protected mice from BA-induced cytotoxicity by increasing expression of hepatic genes involved in BA metabolism and transport [132]. This beneficial role of CAR in the protection against hepatotoxic LCA is likely attributed to the induction of phase I and II enzymes (Cyp3a4, Ugt1a1), as well as efflux transporters (Mrp2), in these cholestatic animal models [52,132,133]. Although PXR and CAR were first identified as xenobiotic receptors, a body of evidence is now surfacing which indicates equally important roles for these receptors in endobiotic homeostasis [74]. As a whole, PXR and CAR appear to cooperate with FXRα for protection against endobiotic and xenobiotic toxicity, making these NRs interesting targets for therapeutical approaches in the treatment of metabolic disorders, although their role beyond being BA/xenobiotic NRs need to be expanded to include their effects on health and diseases, while minimizing adverse diet-drug or drug-drug interactions (DDIs) in patients with chronic diseases.

The occurrence of these unwanted DDIs further complicates the modulation of FXRα, PXR, and CAR in patients affected with cancers, because these NRs are major transcriptional regulators of drug-metabolizing enzymes with clinical consequences that generally decrease therapeutic efficacy, and occasionally increase drug toxicity. Early work of De Gottari et al. suggested that FXRα loss would contribute to tumorigenesis of colorectal cancer in human, and supports the view that BAs are likely carcinogens in human and rodent [134,135]. In this view, mice lacking Fxrα with impaired BA homeostasis spontaneously develop hepatocellular carcinomas, and treatment with the bile salts sequestrant cholestyramine reduces tumor incidence [136]. In addition, target disruption of the Fxrα in the intestine enhances tumor formation rate in mice, and further supports the existence of a causative link between high BA concentrations and gastrointestinal cancer [137]. Therefore, activation of Fxrα by non BA ligands was proposed as an attractive strategy to protect against liver and intestinal carcinogenesis, but the mechanism by which FXRα suppresses carcinogenesis remains to be investigated, since FXRα agonists have also been been reported to enhance chemoresistance in colon and liver cells [138,139]. This mechanism possibly involves increased expression of gene implicated in drug metabolism and transport to inactivate and export the anticancer drug out of expressing cells with an impact on chemotherapy outcome. A similar theme plays out for PXR and CAR, and their crosstalks not only gained increased attention for the treatment of metabolic diseases, but they have also revealed context-specific effects of PXR and CAR activation in treating cancers [94,95,140,141,142]. Owing to the roles of PXR in promoting cell growth, chemoresistance, and malignancy, PXR antagonists are of potential therapeutic interest for malignant diseases [143,144]. Indeed, developing selective natural (ET-743, sulforaphane, coumestrol) and synthetic (ketoconazole and its derivatives) PXR antagonists appears to be a reasonable approach for managing PXR-related adverse DDIs and cancer drug resistance [94]. Regarding CAR, its function is less clear in the maintenance and the development of chemoresistance. On the one hand, PB and TCPOBOP promote the development of hepatocellular carcinoma in rodent models, and it is thus accepted that CAR activators may act as cancer promoters in mice, but not in human, although some reports suggest that chemotherapeutic efficacy may be increased upon CAR activation [145,146]. On the other hand, CAR regulates the expression of efflux pump transporters for eliminating anticancer agents, and is thus suspected to play a major role in the development of chemoresistance [147]. Indeed, before considering human CAR as a therapeutic target for cancer, extensive investigation into the functioning of the receptor in human needs to be performed. In light of these studies, the differential effects of these NRs in different cancer tissues might in part due to different variants that exhibit differential expression, ligand binding affinity, and transcriptional activity. It is also possible that polymorphisms and post-translational modifications contribute to the differential roles of these BA/xenobiotic-activated NRs in human cancers. Therefore, the development of effective and selective compounds for each receptor, based on the signaling pathway and disease they are involved in, is crucial for validating FXRα, PXR, or CAR as clinically-relevant, disease-related therapeutic targets, in order to minimize side-effects and improve efficacy.

## 4. Bile Acid/Xenobiotic Nuclear Receptors in Testis: Protection from Toxicity or Obstacle to Chemotherapy?

Pharmalogical modulations of BA transporters and their regulatory NRs have significantly contributed to the development of novel treatment strategies for several hepatobiliary disorders, including cholestasis, gallstones, and fatty liver diseases. In addition, clinical and experimental data indicate that enhanced bile flow by the secondary BA UCDA might produce other beneficial effects by preventing hypogonadism in patients with long-standing cholestasis and alleviating reproductive system alterations, as shown in male rat with chronic cholestasis induced by permanent common bile duct ligation [148,149]. These positive effects of UCDA suggest that unbalanced BA levels might be deleterious to both endocrine and exocrine functions of the testes, and have warranted numerous studies that have highlighted a significant link between BA metabolism and male fertility disorders. Firstly, feeding CA during pubertal age reduces testosterone concentrations and increases germ cell apoptosis, impairing fertility in adult mice [150,151]. Prevention of germinal cell apoptosis by testosterone indicates that the steroidogenic function of Leydig cells is likely impaired in these mice, in agreement with previous studies which reported that agonist activation of Fxrα reduces testicular steroid metabolism through down-regulation of steroidogenic genes including *Star*, *Cyp11a1,* and *3β-Hsd* [152]. Secondly, BA-sensing NRs including FXRα, PXR, and CAR are expressed in both human and mouse testis [153,154], and a specific panel of conjugated and unconjugated BAs including the BA intermediates dihydroxycholestanoic acid (DHCA) and trihydroxycholestanoic acid (THCA) have been reported in this tissue in mice [155]. The physiological relevance of these BA/xenobiotic, NRs-dependent signaling pathways in the gonadal function and its regulation remain elusive, but the increased number of undifferentiated germinal cells found in conjunction with the extended reproductive longevity in male mice lacking Fxrα underscores the need to address the mechanisms underlying these effects [156]. One would expect that FXRα might also crosstalk with other NRs such as CAR, since impaired spermatogenesis due to the higher germ cell apoptosis observed in Fxrα^-/-^ male mice fed CA-supplemented diet is reversed by co-administration of the CAR agonist TCPOBOP, while the CAR inverse agonist androstanol can reproduce these defects in wild-type mice after two weeks of treatment [155]. Moreover, the recent demonstration that the FXRα antagonist stigmasterol can modulate the adverse events of the common environmental chemical bisphenol A (BPA) on testis development and spermatogenesis is questionable, since natural ligands of FXRα with antagonist properties might dramatically accelerate germ cell loss by external factors, including environmental toxicants, leading to reduced fertility (Figure 3 and [157]).

Additional aspects should also be considered when evaluating the roles of BA and xenobiotic-regulated signaling networks in male fertility outcomes because numerous drug transporters in the testis have been described [160]. Although these protein transporters are best studied in cancer cells, the cellular distribution of efflux and influx pumps in Sertoli and germ cells confers an exclusive microenvironment at the apical compartment near the blood-testis barrier (BTB) for meiosis and postmeiotic spermatid development upon spermiogenesis, with the renewal of undifferentiated spermatogonial stem cells and their mitotic proliferation taking place outside of the BTB at the basal compartment. By regulating the expression of drug transporters, all three NRs (FXRα, PXR, and CAR) can moderate the passage of BAs and BA-derived metabolites, as well as of environmental chemicals across the seminiferous epithelium to safeguard meiosis and spermiogenesis that occur in the apical compartment [160,161]. Therefore, FXRα, PXR, or CAR -specific ligands might potentially modulate the expression of drug transporters, suggesting the direct involvement of these NRs in controlling the entry of endogenous and exogenous molecules (into intratubular Sertoli and germ cells) through uptake transporters, or their elimination throughout efflux pumps. Moreover, BA-induced down-regulation of adhesion proteins like N-cadherin and connexin 43 (CX43) was shown to alter the BTB, leading to its disruption [162], making way for more BAs to access the apical compartment and to induce male reproductive dysfunctions by amplifying germ cell degeneration. As observed for male rat fed high-fat diets [163], it cannot be excluded that BA overload causes persistent alteration in the male germ cell epigenome with an impact on offspring, in accordance with recent studies that have reported generational inheritance of metabolic defects in the offspring of CA-fed fathers [164,165]. Unbiased whole-genome approaches should help to address how BAs code transgenerationally-inherited traits, and complementary studies are still needed to tease out the complex relationship between BAs, their receptors, and male germ cell-derived epigenetic inheritance. This is a main issue to tackle since it would contribute to determining the mechanisms of germline-dependent transmission for epigenetic information, as already noted for other environmental contaminants (bisphenol analogs, phthalates, and organochloride pesticides) that recapitulate some BA-induced injuries by acting as endocrine-disrupting compounds (EDC) and demonstrate a trend of increasing testicular germ cell cancer (TGCC) [166].

In this context, the normal function of the BTB is to protect germinal cells against the infiltration of harmful toxicants that can disrupt spermatogenesis. Also, it can act as an obstacle to chemotherapy by impeding the delivery of chemotherapeutic drugs to the testis by pumping unwanted drugs out to avoid harmful effects. From an oncological view, the regulation of ABC transporters (MDR1, BCRP and MRP1) and OATP membrane transport proteins by BA/xenobiotic-activated NRs at the human BTB might interfere with the ability of systemic chemotherapy to successfully treat germ cell cancers within the testis [160,167,168]. The concentration of BAs and their metabolites should thus be carefully assessed as well in patients undergoing cytotoxic chemotherapy with cholestasis, since accumulation of BAs resulting from impaired bile flow may activate mechanisms of tumor cell resistance to chemotherapy through the regulation of drug-metabolizing enzymes and transporters. Although the increasing incidence of testicular germ cell cancers (TGCC) is, for the most part, still unexplained, exposure to endogenous toxicants and environmental contaminants has been suggested as one possibility, albeit with only limited evidence. In addition, other disorders of male reproductive development such as impaired testosterone production, low spermatozoa count, and infertility in adulthood are also associated with increased risk of TGCC [169,170]. It is thus tempting to speculate about the different forms of clinical presentations of drug-induced liver injury in malignant transformation by prolonged exposure to BAs in the testis with an impact on reproductive functions and chemotherapy outcome.

## 5. Conclusions

The studies described herein depict a comprehensive overview that focuses on the organized actions of BA and xenobiotic sensing NRs in regulating various physiological processes in addition to drug metabolism and disposition. In this context, examples of crosstalks reveal that signaling pathways controlling xenobiotic/drug metabolism are closely embedded within regulatory networks governing BA homeostasis. Understanding the mechanisms that modulate BA synthesis and transport under (patho)physiologic conditions thus represents a very challenging issue for the improvement of therapeutic approaches for liver diseases with imbalanced bile flow. Within enterohepatic circulation, BAs exert numerous functions, like facilitating intestinal lipid absorption, clearance of potentially toxic molecules, and control of energy metabolism. Moreover, the differential roles of FXRα, PXR, and CAR in cancers suggest that several mechanisms may be involved in BA/xenobiotic NRs-mediated tumor growth or chemotherapeutic response. Mechanistic studies will be critical to thoroughly understanding the role of these NRs in tumor progression or suppression, as well as in chemoresistance or chemosensitivity. A comprehensive investigation of these BA and xenobiotic NRs is also need to establish how these three sensing NRs might contribute, in different isoforms and alternative spliced variants, to the regulation of male germ cell proliferation and differentiation with an impact on fertility and TGCC incidence. As a first step, a specific pool of BA intermediates has been characterized in the testis, indicating the possibility that BAs might affect testis development and spermatogenesis, as supported in mice treated with Fxrα agonists which display testis alterations, increased spermatogonial germ cells degeneration, and reduced fertility. In addition, an increasing number of reports indicate that FXRα and both xenosensors PXR and CAR have an extensive presence in testes, suggesting a possible role of these receptors in the protection of germ cells against the adverse effects of toxic BA byproducts or environmental contaminants such as bisphenol analogs, phthalates, and organochloride pesticides. Discriminating the participation of FXRα, PXR, and CAR in reproduction requires consideration of their interactions with specific ligands, as well as their crosstalks in the coordination of endobiotic- and xenobiotic-detoxification systems. Decrypting the contribution of BA and xenobiotic NRs, and their interactions in health and diseases, will advance the development and expand the utilization of targeted small molecules to cure human metabolic disabilities. Still, there are too few studies of this nature on the testis, and more research is needed to better understand how BAs and xenobiotics can impact their development and function.

## Figures and Tables

**Figure 1 ijms-19-03630-f001:**
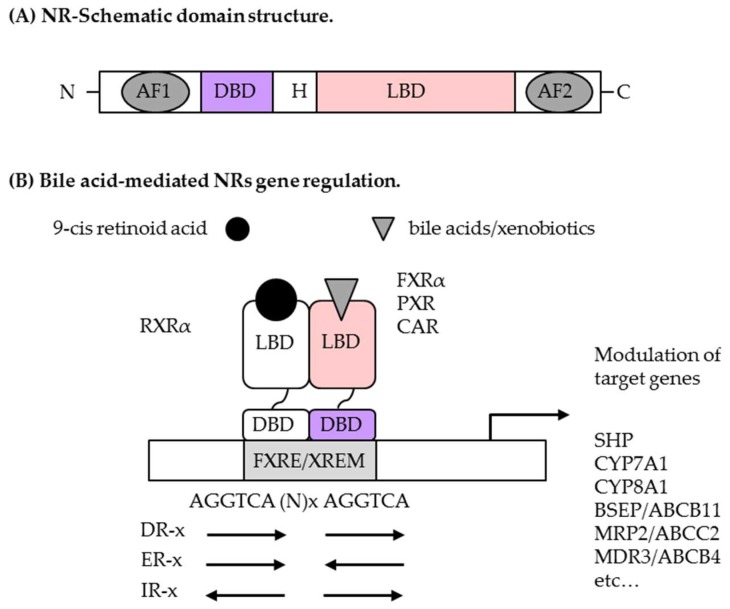
Structural organization and mechanism of action of nuclear receptors. (**A**) Structural organization of hormone nuclear receptors. AF-1, activation function 1; DBD, DNA binding domain; H, hinge region; LBD, ligand binding domain; AF-2, transactivation function 2. (**B**) A current model of NR-mediated gene regulation. Bile acid-activated NRs (FXRα, PXR, CAR) act mainly as heterodimers with retinoid X receptor α (RXRα) to regulate gene transcription. FXRE, FXRα response element; XREM, xenobiotic response element; DR, direct repeat; ER, everted repeat; IR, inverted repeat.

**Figure 2 ijms-19-03630-f002:**
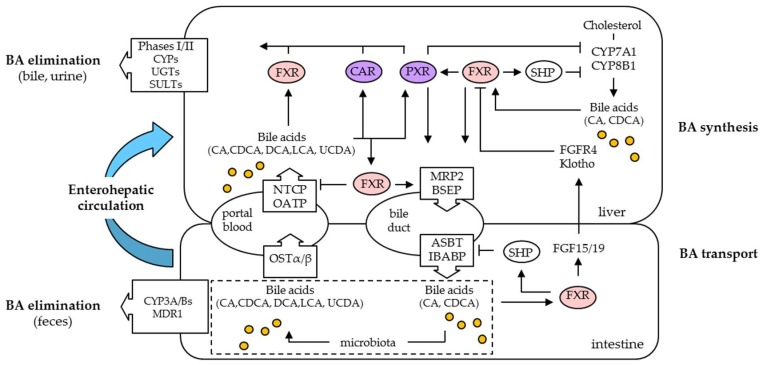
Model of effects of FXRα on bile acid (BA) metabolism. Primary BAs, CA, and CDCA, are generated from cholesterol by liver enzymes, including CYP7A1 and CYP8B1. CA and CDCA are metabolized to DCA and LCA secondary BAs in the intestine. BSEP and MRP2 are localized in the canalicular membrane of hepatocytes and transport BAs into bile. Most BAs are reabsorbed in the intestine and recirculate to the liver through the portal vein in a mechanism called the enterohepatic circulation. BAs that escape reabsorption are excreted in the feces. A portion of secondary BAs enters the enterohepatic circulation from the colon. The transporters ASBT and OSTα/β are involved in BA absorption in the intestine. At the basolateral membrane of hepatocytes, NTCP and OATPs uptake BAs from the portal blood. Activated FXRα, PXR, and CAR induce the expression of phases I and II metabolizing enzymes to stimulate urinary and biliary excretion of BAs.

**Figure 3 ijms-19-03630-f003:**
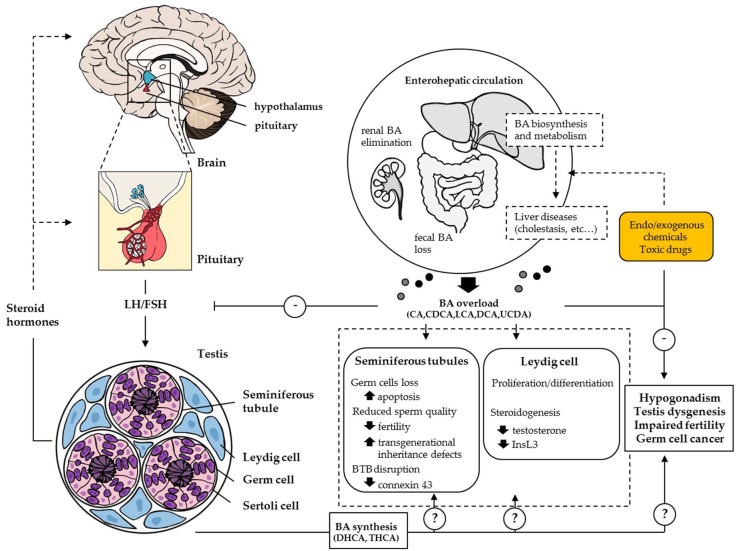
Overview of reproductive disorders linked to disturbance in BA homeostasis. Disturbance in BA homeostasis within the enterohepatic cycle results in hypogonadism with reduced androgens synthesis in the testis. Under normal conditions, BAs that escape the enterohepatic circulation are excreted in urine and feces, but BA overload due to impaired bile flow or xenobiotic/drug-induced liver cytotoxicity can cause a variety of fertility disorders. High BA levels reduce steroidogenesis in Leydig cells and disrupt the BTB, leading to enhanced germ cell degeneration with an impact on reproductive functions. In mouse, BA synthesis also takes place in the testis where germ cells develop from the tubule wall toward the central lumen (modified from [158,159]).

**Table 1 ijms-19-03630-t001:** Chemical structures of the most common bile acids (BAs) and their conjugated forms (with taurine or glycine). The positions of hydroxyl groups, which are responsible for major physicochemical differences among the BA species, are indicated. Cholic acid (CA) and chenodeoxycholic acid (CDCA) are primary BAs. Deoxycholic acid (DCA) and lithocholic acid (LCA) are secondary BAs. The hydrophobicity increases as follows: MCA, UDCA, CA, CDCA, DCA, LCA.

Chemical Structures of the Most Common Bile Acids (BAs)	R1 (3-pos)	R2 (6-pos)	R3 (7-pos)	R4 (12-pos)	Bile Acids and BAs Derivatives
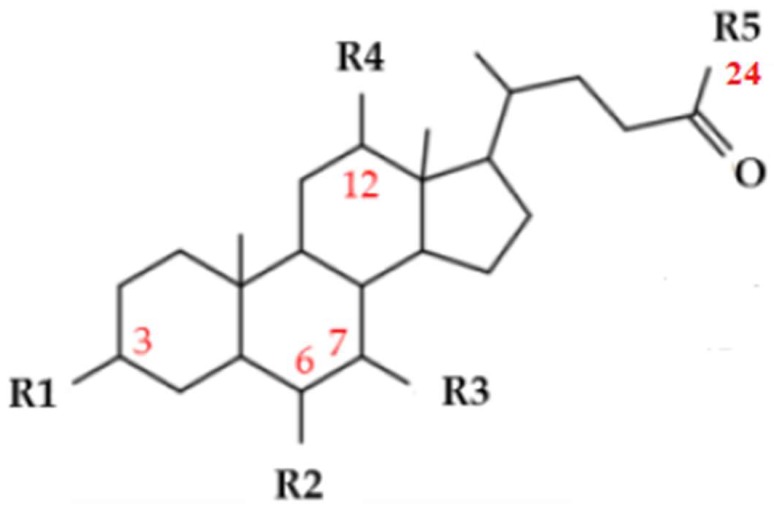	OH	H	OH	OH	Cholic Acid (CA)
	OH	H	OH	H	Chenodeoxycholic Acid (CDCA)
	OH	H	H	OH	Deoxycholic Acid (DCA)
	OH	H	H	H	Lithocholic Acid (LCA)
	OH	OH	OH	H	Muricholic Acid (MCA)
	OH	H	OH	H	Ursodeoxycholic Acid (UDCA)
Free BAs: R5 (24-pos) = OH	C=O	H	H	H	3-Keto LCA
Tauro-conjugated BAs: R5 = NHCH_2_CH_2_SO_3_H	OH	C=O	H	H	6-Keto LCA
Glyco-conjugated BAs: R5 = NHCH_2_CO_2_H	OH	C_2_H_5_	OH	H	6α-Ethyl CDCA (6-ECDCA)

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
