# Peer review of "Nuclear Receptor Metabolism of Bile Acids and Xenobiotics: A Coordinated Detoxification System with Impact on Health and Diseases"

_ijms, 2018, doi:10.3390/ijms19113630_

Round 1

Reviewer 1 Report

In this manuscript, the authors review nuclear receptors that regulate bile acid and xenobiotic metabolism. This is a well written review paper about the nuclear receptors FXR, PXR and CAR. I have only minor comments.

1. Introduction, lines 31-32. Is FXRβ necessary? I don't think that FXRβ (NR1H5) has been intensively studies. Explanation about FXRβ in Section 2.2 is enough.

2. Introduction, line 41. Is only RXRα a heterodimer partner for FXR, PXR and CAR? Please explain about other RXRs, RXRβ and RXRγ.

3. Please check English carefully. For example, line 90, "indicate" may be "indicates".

Author Response

Dear Editors,

We would like to take this opportunity to express our sincere thanks to both reviewers who identified areas of our manuscript that needed corrections or modifications before its acceptance for publication. We hope that our point-by-point response and the corresponding modifications in the manuscript will authorize its revised form to be acceptable for publication in the special issue Molecular Biology of Nuclear Receptors.

Reviewer 1 (remarks to the author):

Point 1: Introduction, lines 31-32. Is FXRβ necessary? I don't think that FXRβ (NR1H5) has been intensively studies. Explanation about FXRβ in Section 2.2 is enough.

Response 1: As requested, the nomination of FXRβ has been removed since a detailed presentation of FXRα and β isoforms is reported in section 2.2 (lines 116-124).

Point 2: Introduction, line 41. Is only RXRα a heterodimer partner for FXR, PXR and CAR? Please explain about other RXRs, RXRβ and RXRγ.

Response 2: This point is now clarified in the revised version of the manuscript (lines 41-45). The text is now stating: "As common (non)permissive heterodomeric partners, retinoid X receptors (RXRα/β/γ, also known as NR2B1/NR2B2/NR2B3) are required for high affinity binding to DNA and despite the fact that RXRα remains the predominant isotype in many tissues, all three RXR subtypes generally make these NRs competent for chromatin binding and transcriptional regulation of target gene expression in concert with corepressor complexes or recruited co-activator proteins (Figure 1B and [4, 5])."

Point 3: Please check English carefully. For example, line 90, "indicate" may be "indicates".

Response 3: The manuscript has been carefully checked by the authors and several changes have been made in addition to "indicates" (line 90).

Reviewer 2 Report

this article covers current and very interesting topics for biochemists, pharmacologists, etc. the article is very well structured, easy to read, and is well illustrated.

Minor:

Authors could insert the structure of bile acids: CA, CDCA, DCA, LCA, MCA, 6-keto-LCA, 6-ECDCA.

Author Response

Dear Editors,

We would like to take this opportunity to express our sincere thanks to both reviewers who identified areas of our manuscript that needed corrections or modifications before its acceptance for publication. We hope that our point-by-point response and the corresponding modifications in the manuscript will authorize its revised form to be acceptable for publication in the special issue Molecular Biology of Nuclear Receptors.

Reviewer 2 (remarks to the author):

Point 1: Authors could insert the structure of bile acids: CA, CDCA, DCA, LCA, MCA, 6-keto-LCA, 6-ECDCA.

Response 1: A table has been included in the revised version as requested. This table presents the chemical structures of the most common bile acids as well as their conjugated forms with taurine and glycine. This should improve the quality of the review and quickly help readers.